# Molecular Characterization and Expression Pattern of Paramyosin in Larvae and Adults of Yesso Scallop

**DOI:** 10.3390/biology11030453

**Published:** 2022-03-16

**Authors:** Yumin Yang, Dan Zhao, Liqing Zhou, Tianshi Zhang, Zhihong Liu, Biao Wu, Tao Yu, Yanxin Zheng, Xiujun Sun

**Affiliations:** 1Fisheries College, Zhejiang Ocean University, Zhoushan 316022, China; ym1290576128@163.com; 2Laboratory for Marine Fisheries Science and Food Production Processes, Pilot National Laboratory for Marine Science and Technology, Yellow Sea Fisheries Research Institute, Chinese Academy of Fishery Sciences, Qingdao 266071, China; zdan12072021@163.com (D.Z.); zhoulq@ysfri.ac.cn (L.Z.); zhangts@ysfri.ac.cn (T.Z.); liuzh@ysfri.ac.cn (Z.L.); wubiao@ysfri.ac.cn (B.W.); 3Changdao Enhancement and Experiment Station, Chinese Academy of Fishery Sciences, Yantai 265800, China; cdyutao@126.com (T.Y.); zhengyanxin1989@163.com (Y.Z.)

**Keywords:** scallops, smooth adductor muscle, paramyosin, gene expression, whole-mount in situ hybridization

## Abstract

**Simple Summary:**

Paramyosin is an important myofibrillar protein in smooth muscle in molluscs that is not present in vertebrate muscles. This study characterized its sequence feature and expression patterns in Yesso scallop *Patinopecten yessoensis* and revealed the unique phosphorylation sites in scallops. The mRNA and protein expression of paramyosin was mainly found in foot and smooth adductor muscle. At late larval stages, strong paramyosin mRNA signals were detected in the symmetric positions of anterior and posterior adductor muscles. The present findings support that paramyosin may serve as the most important component of smooth muscle assembly during muscle development and catch regulation in scallops.

**Abstract:**

Paramyosin is an important myofibrillar protein in molluscan smooth muscle. The full-length cDNA encoding paramyosin has been identified from Yesso scallop *Patinopecten yessoensis*. The length of paramyosin molecule has been found to be 3715 bp, which contains an open reading frame (ORF) of 2805 bp for 934 amino acid residues. Characterization of *P. yessoensis* paramyosin reveals the typical structural feature of coiled-coil protein, including six α-helix (α1-α6) and one coil (η) structures. Multiple phosphorylation sites have been predicted at the N-terminus of paramyosin, representing the unique phosphorylation sites in scallops. The highest levels of mRNA and protein expression of paramyosin have been found in foot and the smooth adductor muscle. According to whole-mount in situ hybridization (WISH), strong paramyosin mRNA signals were detected in the symmetric positions of anterior and posterior adductor muscles at late larval stages. These findings support that paramyosin may serve as the most important components for myogenesis and catch regulation in scallops. The present findings will not only help uncover the potential function of myofibrillar proteins in molluscs but also provide molecular evidence to infer evolutionary relationships among invertebrates.

## 1. Introduction

For both of vertebrates and invertebrates, muscles play a central role in body mobility, stability maintenance, and metabolic function [1,2]. Myofibers, also known as muscle cells, can generate the required force for muscle function and body locomotion. Myofibers are generally made up of thick and thin filaments, which are composed of accessory structural proteins, such as myosin, actin, and troponin [2,3]. The accurate assembly of filament proteins is essential for muscle development and function [4,5]. Although vertebrate and invertebrate muscles share many fundamentally similarities, invertebrate muscles have several differences from vertebrate muscles [4]. For instance, invertebrate muscles usually have extraordinary diversity of muscle-specific protein isoforms [1,3,4,6,7]. Furthermore, paramyosin-rich thick fibers, showing the large and variable diameters, are only found in invertebrate muscles [4]. The unusual structure of thick filaments in invertebrates makes the smooth muscle possess a unique property, known as “catch”, which maintains contraction with a low cost of energy [8,9].

In the past few decades, a lot of work has been done on invertebrate muscles, including muscle-specific genes and proteins, filament structure, and the molecular basis of contraction and its regulation (reviewed by 3,4). The accurate assembly and folding of myofibrillar proteins are necessary for muscle development and function [10,11]. Studying invertebrate muscle genes and proteins will provide key information on muscle protein diversity, genetic mechanisms, and phylogenetic relationships [3,4].

Paramyosin is one of the most unique muscle proteins in invertebrates, which is not present in vertebrates. Paramyosin is usually present in large and small diameter thick filaments in a variety of invertebrate species, especially in molluscs [3]. The content of paramyosin in myofibrillar proteins varies among species and muscle types, ranging from 3% (striated muscle) to 65% (smooth muscle) in molluscs [12,13]. As suggested, the paramyosin rods may form a core of thick filaments, which can affect the orientation of the myosin molecules [14]. As indicated, the generated force of muscle contraction in invertebrate muscles has a great correlation with the length of thick filaments, which is determined by the paramyosin content [6,15]. One possible explanation is that paramyosin:paramyosin binding may be stronger than myosin:myosin tail binding, which may serve as a structural element in very large diameter filaments [4]. More and more evidence indicates that paramyosin phosphorylation is essential for muscle force generation in invertebrates, which is probably involved in catch contractions [15,16,17,18,19,20]. Indeed, the interaction of entire rod region of myosin with paramyosin core might be modulated by the phosphorylation of paramyosin during the catch state [21]. However, most of these studies are related to the structural aspects of paramyosin in molluscan smooth muscles. Comparatively little work has been done on the molecular characterization, expression pattern, and phosphorylation of paramyosin in molluscs.

As indicated in recent studies, paramyosin might be involved in immune response and protective immunity in invertebrates, which have been identified as potential vaccine candidates for parasite control [22,23,24]. Molluscs have been served as a very important seafood resource for human consumption due to their delicious taste and nutrient content in muscle tissues [25]. For instance, scallops are favored by global consumers for the gastronomic delights of scallop adductor muscles [26]. The increasing consumption of seafood has been accompanied by rapid growth in the prevalence of seafood allergies. Paramyosin has been recently revealed as a novel allergen in molluscan shellfish [27,28,29]. In our previous studies, paramyosin has been identified in both of the smooth and striated adductor muscles in scallops [30,31,32]. However, the gene structure and expression pattern of paramyosin has not been well studied yet in scallops or in other molluscs.

For molluscs, paramyosin is also potentially served as a mollusc allergen and potential vaccine. However, most of these potential functions remain poorly understood. To address these questions, it is important to characterize the gene structure and gene expression pattern in molluscs. Here, the full-length cDNA of paramyosin was isolated and characterized from an economically important species, Yesso scallop *Patinopecten yessoensis*. The temporal and spatial expression patterns of paramyosin in scallop muscle tissues were determined by qPCR, immunohistochemistry staining, and whole-mount in situ hybridization. Our present findings enhance our understanding of the molecular basis of muscle protein diversity, catch regulation, and phylogenetic relationships, in molluscs.

## 2. Materials and Methods

### 2.1. Experimental Animals and Sample Collection

The healthy adults of scallop *P. yessoensis* (average weight: 73.34 ± 20.5 g) were purchased from Nanshan aquatic market (Qingdao, China). They were cultured in 20-L seawater tanks at 15–16 °C for one week before processing. The tissues (smooth and striated adductor muscles, hepatopancreas, gonad, mantle tissue, foot, and gill) were immediately frozen in liquid nitrogen and stored at −80 °C for RNA extraction. Larval samples were collected by 45 μm mesh in Haiyi shellfish hatchery, Yantai, China. For whole-mount in situ hybridization, larvae were fixed using 4% paraformaldehyde (PFA) for 2–4 h at 4 °C and stored at −20 °C long term.

### 2.2. RNA Extraction and cDNA Synthesis

Trizol method (Solarbio, Beijing, China) was used to extract total RNA from the scallops, including smooth and striated adductor muscles, hepatopancreas, gonad, mantle tissue, foot, and gill. The quality and purity of RNA products were assessed using the Nanodrop 2000 spectrophotometer (Thermo Fisher Scientific, Waltham, MA, USA) and 1% agarose gel electrophoresis. The cDNA of paramyosin was amplified using HiScript III RT SuperMix for qPCR kit (Vazyme, Nanjing, China) according to the manufactures’ instructions. The full-length paramyosin cDNA of *P. yessoensis* was obtained according to the previous study [31]. The primers used for the full-length amplification of paramyosin cDNA were listed in Table 1. The obtained fragment sequences were assembled to the full length of paramyosin cDNA using the program SeqMan (DNASTAR Inc., Madison, WI, USA).

### 2.3. Bioinformatics Analysis

The open reading frame of paramyosin was determined using the ORF Finder (https://www.ncbi.nlm.nih.gov/orffinder/, accessed on 15 September 2020). The functional conserved domain and coiled coil regions were predicted by SMART (http://smart.embl.de/, accessed on 15 September 2020) and HMMER web server (https://www.ebi.ac.uk/Tools/hmmer/, accessed on 15 September 2020). The ProtParam tool in ExPASy (https://web.expasy.org/protparam/, accessed on 15 September 2020) was used to predict the molecular weight (Mw) and theoretical isoelectric point (pI) of the protein. The signal peptide of Py-Pmy protein was predicted using SignalP 5.0 Server [33]. The potential transmembrane region was analyzed using TMHMM 2.0 (http://www.cbs.dtu.dk/services/TMHMM/, accessed on 15 September 2020). The potential phosphorylation sites for paramyosin were predicted by NetPhos 3.1 Server (https://services.healthtech.dtu.dk/service.php?NetPhos-3.1, accessed on 5 November 2021). The three-dimensional structure of paramyosin protein was predicted by Swiss model (https://swissmodel.expasy.org/interactive, accessed on 15 September 2020). The amino acid alignment and secondary structure prediction for paramyosin were performed and visualized by ESPript 3.0 [34]. The phylogenetic tree was constructed using the Neighbor–Joining (NJ) method with MEGA 7.0 software.

### 2.4. Gene Expression Analysis by qRT-PCR

All gene specific primers were designed by Primer 3 (http://bioinfo.ut.ee/primer3-0.4.0/; accessed on 11 February 2020) and listed in Table 1. According to the efficiency of PCR amplification, ubiquitin (UBQ) was selected as the internal control for qPCR analysis [31]. The amplification efficiency of each primer pair was calculated by using a 5-fold serial dilution of cDNAs. All the qPCR analysis was performed using a StepOnePlus Real-Time PCR Detection System (Applied Biosystems, Waltham, MA, USA) in a 20 μL reaction system, including master mix, cDNA template, forward and reverse primers, and ROX reference dye I (Vazyme, Nanjing, China). The qPCR reaction was performed under the following conditions: 95 °C for 30 s, followed by 40 cycles (95 °C for 10 s, 60 °C for 30 s). The relative expression levels of the target genes were calculated with the 2−ΔΔCt method. The gene expression data were analyzed by one-way ANOVA with least squares (LSD) in SPSS17.0, and the level of significant difference was set at *p* < 0.05.

### 2.5. The Whole-Mount In Situ Hybridization (WISH)

The anti-sense and sense RNA probes for whole-mount in situ hybridization (WISH) were synthesized using in vitro transcription T7 kit for siRNA synthesis (TaKaRa, Kusatsu, Japan) according to the previous study [31]. Briefly, the anti-sense RNA probe of paramyosin (611 bp) was amplified using the specific primers for WISH (wish_Paramyosin_R and Paramyosin_F). In contrast, the sense RNA probe was synthesized by another pair of primers (wish_Paramyosin_F and Paramyosin_R), which were served as the negative control. For larval samples, they were processed by a series of ethanol gradients and replaced with phosphate buffered saline (PBST) containing 0.1% Tween-20. The samples were then digested by protease K at 37 °C for 30 min. Pre-hybridization was carried out in pre-hybridization buffer (5 × SSC, 50% formamide, 100 μg/mL yeast t-RNA, 1.5% blocking reagent, 5 mM EDTA, and 0.1% Tween-20) for 4 h at 65 °C. Subsequently, they were hybridized with 25 ng/mL RNA probes overnight at 65 °C. Washing steps were carried out in MABT (0.1% Tween-20, 150 mM sodium chloride, and 100 mM maleic acid) and alkaline Tris-buffer. The samples were then incubated with NBT/BCIP solution and terminated by 4% PFA in darkness. Finally, the hybridization signals were observed using the digital camera on Leica DM 4000b.

### 2.6. Immunochemistry Staining in Adult Tissues

For immunochemistry staining, the tissues were preserved in the general tissue fixative (Servicebio, Wuhan, China), including smooth and striated adductor muscles, hepatopancreas, gonad, mantle tissue, foot, and gill. The fixed samples were then treated by routine paraffin sectioning and Haematoxilin/Eosin staining. Briefly, the samples were sequentially dehydrated in 80%, 95%, and 100% ethanol and xylene. Subsequently, the samples were then embedded in paraffin wax and sectioned with rotary microtome (Leica, Germany). After dewaxing and rehydration, antigens retrieval was performed in 0.05mol/L carbonate and incubated at 4 °C overnight. After washing with PBST (PBS containing 0.05% Tween-20), the slides were blocked with 5% skim milk at 37 °C for one hour and incubated with the developed primary antibody for paramyosin (diluted 1:1000). The present study raised the polyclonal antibody in rabbits against a synthetic peptide corresponding to a unique region of *P. yessoensis* paramyosin, with no cross-reactive with myosin. The polyclonal antibody was developed in the laboratory of Antibody Service Department of Shanghai Sangon (China), under the Regulations of Hubei Province on the Administration of Experimental Animals (License key: SCXK2019-0023). The following samples were then incubated with the goat-anti-rabbit secondary antibody (Servicebio, Wuhan, China) at 37 °C for 45 min. After washing with PBST, it was followed by the treatment of 100 μL sulfuric acid (2 mol/L) to stop the immunoreaction reaction. The images were captured using the digital camera on Leica DM 4000b.

## 3. Results

### 3.1. Cloning and Characterization of Paramyosin

The full-length sequence of paramyosin in *P. yessoensis* was obtained and submitted to Genbank (Accession number. OL331027). The total length of the assembled sequence was 3715 bp, including a 5′-UTR (untranslated region) of 91 bp, an ORF (Open Reading Frame) of 2805 bp, and a 3′-UTR of 819 bp (Figure 1). The canonical start (ATG) and stop (TAA) codons were detected in the ORF, which encoded a polypeptide of 934 amino acids, having the molecular weight of 107.54 kDa and PI of 5.62. No signal peptide nor transmembrane structure was detected in paramyosin from *P. yessoensis*.

### 3.2. Protein Structure and Phosphorylation Prediction

Sequence analysis revealed that the paramyosin protein had one conserved motif known as myosin tail, ranging from the position of 55 to 921 (Figure 2). Among these 934 amino acid sequences, nine coiled-coil regions were predicted by HMMER web server using profile hidden Markov Models (Figure 2). The largest region was located at the position of 368 to 639, while the smallest region was located at the position of 865 to 885.

For phosphorylation sites, there were a total of 96 positive predictions (32 known kinases and 64 non-specific prediction of kinase) at the threshold of 0.5 using NetPhos-3.1 (Figure 3). The most frequent phosphorylation type was Serine (S), which was located in 61 different positions, followed by 25 Threonine (T) and 10 Tyrosine (Y). Among these 32 known kinases, PKC (protein kinase C) was found to have the highest frequency of occurrence (18/32), especially at the N-terminus and C-terminus (Table 2). The other types of phosphorylation sites were also found in the paramyosin protein, including four cdc2, three PKA (protein kinase A), three CKII (casein kinase II), two DNAPK (DNA Protein Kinase), and two EGFR (Epidermal growth factor receptor).

### 3.3. Multiple Sequence Alignments and Phylogenetic Analysis

Sequence alignments of the amino acid sequence for paramyosin among six molluscs were showed in Figure 4. The moderate degree of sequence similarity was found according to the multiple alignment. Compared with *P. yessoensis*, the highest similarity level (92.83%) was found in *Pecten maximus*, followed by *Crassostrea gigas* (72.77%), *C. virginica* (67.84%), *Biomphalaria glabrata* (63.67%), and *Pomacea canaliculate* (61.55%). The paramyosin of *P. yessoensis* was predicted to be composed of six α-helix (α1-α6) and one coil (η) structures (Figure 4). The highly conserved sequences were mainly found in the α-helix structures, while the most variable sequences were detected in those non-helix area, especially at the N-terminal. Notably, the paramyosin sequences of two scallops (*P. yessoensis* and *P. maximus*) were remarkably different to those in other species at the N-terminal. In the N-terminal region, more than 50 amino acids appeared to be the unique sequences of the two scallops. According to the phosphorylation prediction, nine sites were speculated to be phosphorylated in this region, including seven PKC and two cdc2 sites.

The phylogenetic analysis of paramyosin indicated that molluscan species were clustered into three main branches: Bivalvia, Gastropoda, and Cephalopoda (Figure 5). As displayed, scallops, oysters, and mussels formed together into the branch of bivalves, independently from the gastropod animals, such as abalone *Haliotis discus hannai*), marine snail (*Aplysia californica*), and rapa whelk (*Rapana venosa*). All cephalopods clustered together and formed an independent branch with 100% confidence.

### 3.4. Tissue-Specific Expression Pattern of Paramyosin

The tissue-specific expression pattern of paramyosin was analyzed by qPCR in different tissues (Figure 6). The highest expression level of mRNA was observed in foot, followed by the smooth adductor muscle. The expression level of smooth adductor muscle was significantly higher than that in the striated part (*p* < 0.01). No significant difference was found between mantle and the striated adductor muscle. The lowest expression levels were detected in gills, hepatopancreas, and gonad.

The immunohistochemistry staining results revealed a tissue-specific pattern of protein expression. The strong positive signals were consistently detected in the smooth adductor muscle, as demonstrated in its transverse and vertical sections (Figure 7A,B). The uneven expression of paramyosin occurred in different smooth fibers, suggesting the alternative protein isoforms in the smooth adductor muscle (Figure 7A). Compared with the smooth muscle, the striated muscle had a relatively lower expression of paramyosin across the striated fibers (Figure 7C). For mantle, the positive signals of paramyosin were almost found at the outer layer of the mantle tissue (Figure 7D). For foot, paramyosin was mainly detected in the circular layer of foot muscle (Figure 7E). For gill, the positive signals were observed in the major plica at the edge of gill tissue (Figure 7F).

### 3.5. The Developmental Expression of Paramyosin at Different Stages

The spatial expression of paramyosin mRNA at different stages was determined by whole-mount in situ hybridization (ISH; Figure 8). No positive signal was found at the multicellular, blastula, and trochophore stages (Figure 8A–C). Initially, the strong positive signal was mainly found at the position of anterior adductor muscle in D-veliger larvae. In contrast, the weak signal was detected in the surrounding area of velum retractor muscles at D-veliger stage (Figure 8D). Until the umbo stage, another positive signal was also found at the position of posterior adductor muscle, coinciding with that in the anterior adductor muscle (Figure 8E). Similarly, the symmetric distribution of positive signals in two adductor muscles was also observed at the spat stage (Figure 8F).

## 4. Discussion

### 4.1. Sequence Feature of Paramyosin in Molluscs

Paramyosin is found within the core of molluscan thick filaments, e.g., scallop smooth adductor muscles [26]. As previous studies indicated, paramyosin-rich thick filaments in scallops are usually large but variable in size [18]. Paramyosin molecules are known as invertebrate-specific coiled-coil proteins [18,35,36,37,38]. In this study, the typical structural feature of coiled-coil regions in paramyosin has been identified by means of a gliding window of 28 residues in *P. yessoensis*. This coiled-coil structure of paramyosin shares a feature with the heptad repeat in 28-residue repeat zones, as evidenced by paramyosin from Pacific oyster *C. gigas* [7]. In this study, we consistently discover the potential coiled-coil regions, homologous to the rod portion of myosin heavy chains, in the functional domain (myosin tail) of paramyosin. For invertebrate muscles, myosin and paramyosin may interact with each other and co-assemble to form highly ordered thick filaments [7,39,40]. The possible interaction of paramyosin with myosin may impair the association of the myosin heads with F-actin, suggesting a relaxing mechanism within invertebrate muscles [41]. The present study has revealed the same conserved domain in scallop paramyosin, which suggests the potential interaction of paramyosin with myosin during muscle structural assembly, especially for the smooth muscle fibers.

According to the sequence alignment, more than 50 amino acids appear to be the unique sequence of the two scallops in the N-terminal region (Figure 4). However, they are not located in the predicted secondary structure, suggesting the unique sequence may not have any impact on its protein structure and function. In contrast, we have identified 18 phosphorylation sites (16 Serine and 2 Threonine) in the unique sequence (Figure 3). The unique isoform of paramyosin in scallops may be generated by alternative RNA splicing from the same gene in oysters and scallops [30].

### 4.2. Evolution of Paramyosin in Protostomes

Paramyosin is a good marker protein to estimate the diversity and evolution of muscle contractile apparatus in deuterostomes [42,43]. As indicated, paramyosin is widely distributed in protostomes, while its distribution in deuterostomes is limited in echinoderm and hemichordate. For instance, paramyosin has been found highly expressed in the smooth muscle of sea cucumber, sea urchin, and acorn worm [43,44,45]. However, no paramyosin has been identified in either the striated or smooth muscles from vertebrates. As one of the most closely related invertebrates to vertebrates, amphioxus shows the expression of paramyosin in the notochordal lamellae, suggesting the evolutionary transition from invertebrates to vertebrates [46,47]. In the present study, we provide the molecular phylogenetic evidence for evolutionary relationships among invertebrates, suggesting the independent evolutionary origin of paramyosin between arthropods and molluscs.

### 4.3. Tissue and Developmental Expression of Paramyosin in Molluscs

Paramyosin is the muscle protein found in a number of invertebrates, including insects, arthropods, molluscs, and echinoderms [7,48,49,50]. As indicated, paramyosin is absent from chordates, but it has been detected in sea lily muscle as in muscles of the other echinoderms [51]. In *Drosophila*, paramyosin is proved to be important for myoblast fusion, myofibril assembly, and muscle contraction for flight function [50,52]. Recently, the proteomic identification of paramyosin in octopus and jumbo squid helps one understand its physicochemical property on the textural behavior of cephalopod muscle [49].

The thick filaments containing paramyosin are always longer and larger in diameter than the filaments lacking this core protein [8]. Compared with shorter fibers, longer fibers can interact with the surrounding fibers in a greater tension [6]. For molluscan smooth muscles, paramyosin may provide the basis for protein assembly of smooth muscle fibers in maintaining the rigid network and formation of interfilament connections [8,9]. For molluscs, paramyosin is known to be the most important structural component of smooth muscle fibers, accounting for more than 80% of the thick filament mass [15,53]. In contrast to the smooth fibers, the striated fibers usually have a relatively low content of paramyosin (less than 7% by weight) [8,15,18]. SDS-PAGE analysis reveals a significantly higher content of paramyosin in the smooth adductor muscle (31%) than that in the striated muscle (11%) [54]. Characterization of paramyosin in the smooth muscle of worm suggests that paramyosin binds to myosin for the induced formation of aggregates of myosin [43]. As indicated, paramyosin has been identified to be preferentially expressed in the smooth adductor muscle of scallops, which is supported by the association analysis of proteomic and transcriptomic data [30]. In the present study, the differential mRNA and protein expression of paramyosin between the striated and smooth adductor muscles has been revealed by tissue-specific expression pattern using qPCR and immunohistochemistry staining. Similarly, mRNA encoding paramyosin is most abundant in *Mytilus* muscle, such as retractor and adductor muscles [55]. In this study, the immunohistochemistry staining shows that the positive signals are mainly located in sail-like part of the outer mantle edge, circular layer of foot muscle, and gill lobules edge. Together with previous studies, the present results further shed light on the fact that scallop paramyosin is potentially involved in smooth muscle contraction in muscle-related tissues.

For the spatial expression patterns of paramyosin mRNA expression, we detect no positive signal during the early stages, including multicellular, blastula, and trochophore. In contrast, the specific signals of paramyosin have been detected at the gastrula and trochophore stages of Pacific oyster *C. gigas* [7]. The differences may be associated with the different isoforms of paramyosin between scallops and oysters, which is supported by the unique sequence of paramyosin in scallops according to the alignment results (Figure 4). For *C. gigas*, mRNA expression of paramyosin after D-shaped larvae is not limited in the adductor muscles but also found in larval velum retractors. Consistently, we also detect the weak signals of mRNA expression of paramyosin in the surrounding area of velum retractor muscles. As indicated, the musculature of *P. yessoensis* develops into a well-organized muscle system at the veliger stage [56]. In the present study, mRNA expression of paramyosin has been found at the positions of anterior and posterior adductor muscles. For oysters and scallops, the similar spatial distribution of paramyosin indicates that paramyosin may serve as an important structural protein for the formation of well-organized muscle system of veliger larvae across molluscs [57]. For bivalves, the temporal-spatial expression of paramyosin in well-organized muscle system may be responsible for larval swimming and feeding during the planktonic period [7]. Another possible explanation is that paramyosin expression is critically important for protein accumulation until the appearance of the well-organized muscle system [58]. It is therefore suggested that muscle protein accumulation during larval development may participate in reorganization of nonmuscle cytoskeleton into the specialized muscle structure in larvae.

### 4.4. Implication for Catch Mechanism in Molluscs

For vertebrate smooth muscles, caldesmon can crosslink actin and myosin, which may contribute to the latch state [59]. A similar actin-myosin interaction may be involved in the catch mechanism usually found in molluscan smooth muscles [26,60]. Despite this, there is a remarkable difference between catch and latch due to the lack of paramyosin in vertebrate smooth muscles. In the present study, the identification and characterization of paramyosin from *P. yessoensis* highlights the molecular components of catch regulation in scallops. Notably, scallop paramyosin has been predicted to have multiple phosphorylation sites, especially at the N-terminus. The present findings are consistent with a greater degree of paramyosin phosphorylation within the smooth muscle, indicating that paramyosin phosphorylation is potentially involved in the regulation of catch, especially at a slightly acidic pH [9,16].

In *Drosophila melanogaster*, paramyosin phosphorylation is essential for flight muscle stiffness and power generation since phosphorylation sites may reinforce interactions between myosin rod domains, enhance thick filament connections, and stabilize thick filament interactions [19]. In our previous study, identification of phosphorylation sites in paramyosin provided direct evidence of paramyosin phosphorylation in relation to meat quality of scallops [20]. In the present study, multiple phosphorylation sites have been identified at the N-terminus of paramyosin in the smooth adductor muscle, indicating the unique and conserved amino acid sequence in scallops. The phosphorylation sites of paramyosin may have an influence on the interaction of paramyosin-paramyosin or paramyosin-myosin in the smooth adductor muscle [7,16,21]. Protein kinases catalyze the phosphorylation of proteins, which may be involved in many cellular processes, including smooth muscle contraction [61,62]. As indicated, 5-HT treatment can improve the capacity of paramyosin phosphorylation by two to four times, while dephosphorylation of paramyosin can cause the release of catch in molluscs [9,16,17].

In addition to molluscs, a similar function of paramyosin phosphorylation has been found in other invertebrates. For instance, paramyosin phosphorylation possibly plays a role in the regulation of muscle contraction during flight in *Drosophila* [19,52]. The phosphorylation of paramyosin is associated with more aggregated and more concentrated filaments pelleting in the same gradient in *C. elegans* [63]. In addition, paramyosin phosphorylation is probably responsible for the generation of different isoforms, which are located in a large variety of muscle types in *Schistosoma mansoni* [64]. In the present study, the identified phosphorylation sites from scallop paramyosin will provide the basis for comparative studies and functional analysis among molluscs. The abundance of PKC in paramyosin phosphorylation sites suggests that paramyosin phosphorylation may be modulated by protein kinase C in the smooth adductor muscle of scallops. Together with the previous findings, the present results may imply the involvement of paramyosin phosphorylation not only in the regulation of catch but also in myofibril assembly and isoform diversity in scallops.

## 5. Conclusions

In conclusion, characterization of the paramyosin gene in *P. yessoensis* reveals dozens of unique phosphorylation sites in scallops. The mRNA and protein expression of paramyosin was mainly found in foot and smooth adductor muscle. At late larval stages, the strong positive mRNA signals have been detected at the symmetric positions of anterior and posterior adductors. The present findings support that paramyosin may serve as the most important component of smooth muscles during muscle development and catch regulation in scallops. The results not only provide useful information for uncovering muscle structure and function in molluscs but also hint at the molecular evidence for evolutionary relationships among invertebrates.

## Figures and Tables

**Figure 1 biology-11-00453-f001:**
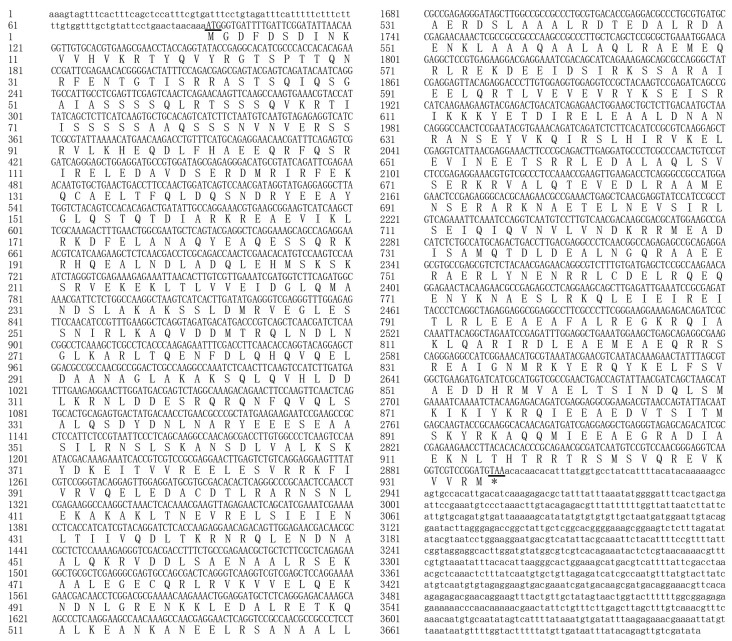
The complete cDNA sequences encoding paramyosin of *Patinopecten yessoensis* and its deduced amino acid sequences. The canonical start (ATG) and stop (TAA) codons are shown in underlines.

**Figure 2 biology-11-00453-f002:**
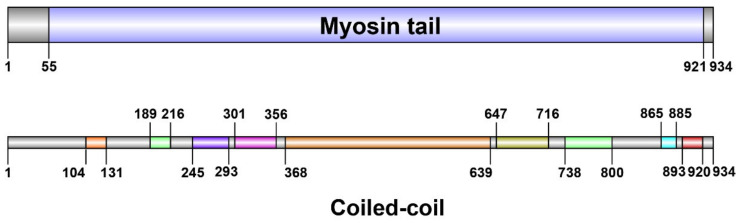
The functional domain and coiled-coil regions predicted by HMMER.

**Figure 3 biology-11-00453-f003:**
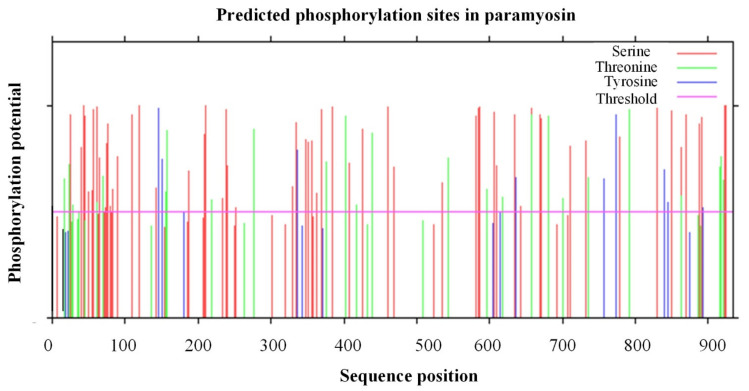
All the potential phosphorylation sites identified in paramyosin of *P. yessoensis* using NetPhos 3.1.

**Figure 4 biology-11-00453-f004:**
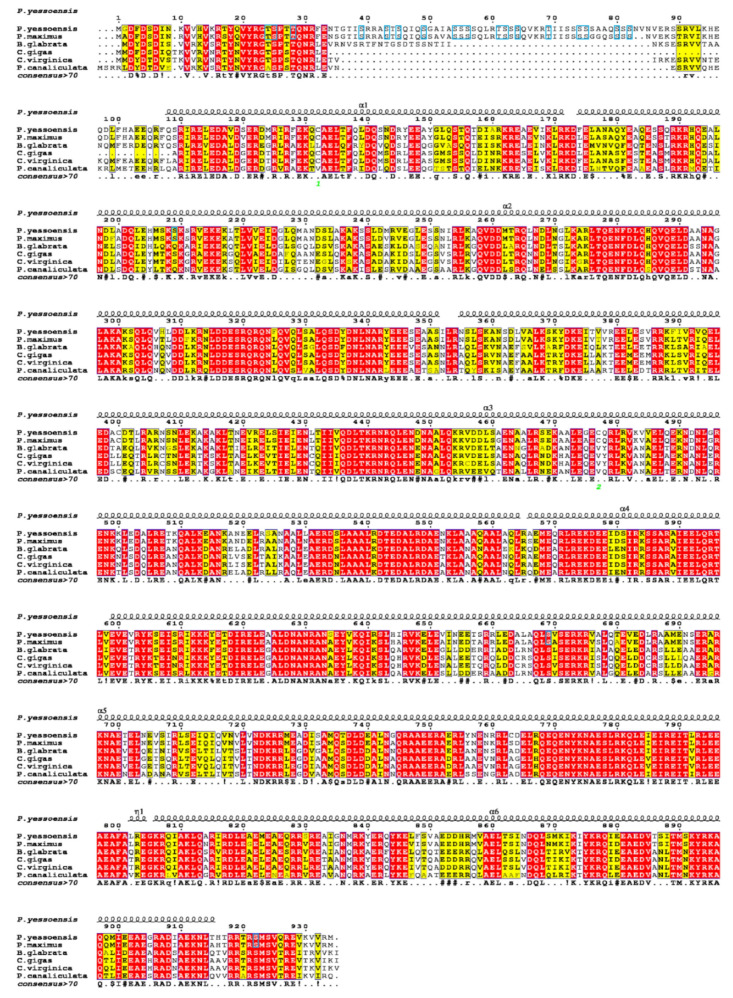
Sequence alignment for amino acid sequences of paramyosin among different molluscan species. The predicted secondary structure of paramyosin is shown at the top. α: α-helix; η: coil. The same phosphorylation sites in *P. yessoensis* and *P. maximus* are marked with blue boxes.

**Figure 5 biology-11-00453-f005:**
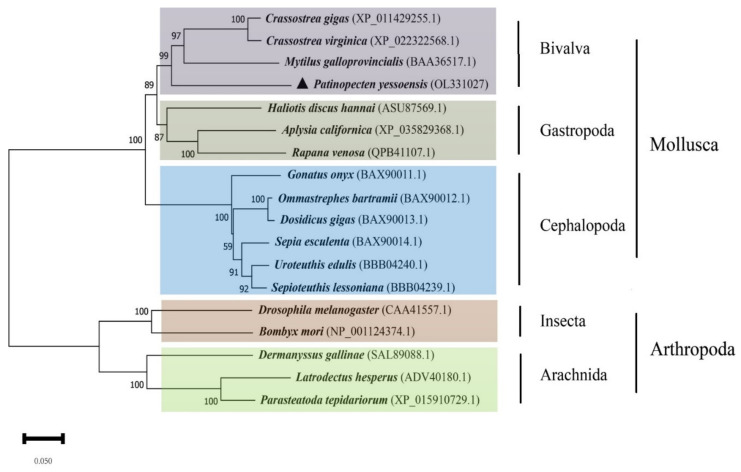
A phylogenetic analysis of paramyosin was constructed based on 18 different invertebrate species. The names of species, and their accession numbers, are indicated in the phylogenetic tree.

**Figure 6 biology-11-00453-f006:**
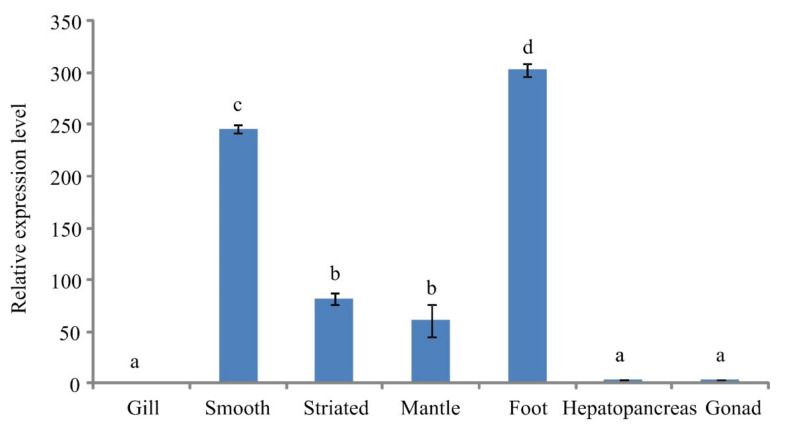
The quantitative expression of paramyosin mRNA among different tissues. Smooth, smooth adductor muscle; striated, striated adductor muscle. Different letters (a, b, c, d) indicate significant differences (*p* < 0.05), while the same letter represents no significant difference.

**Figure 7 biology-11-00453-f007:**
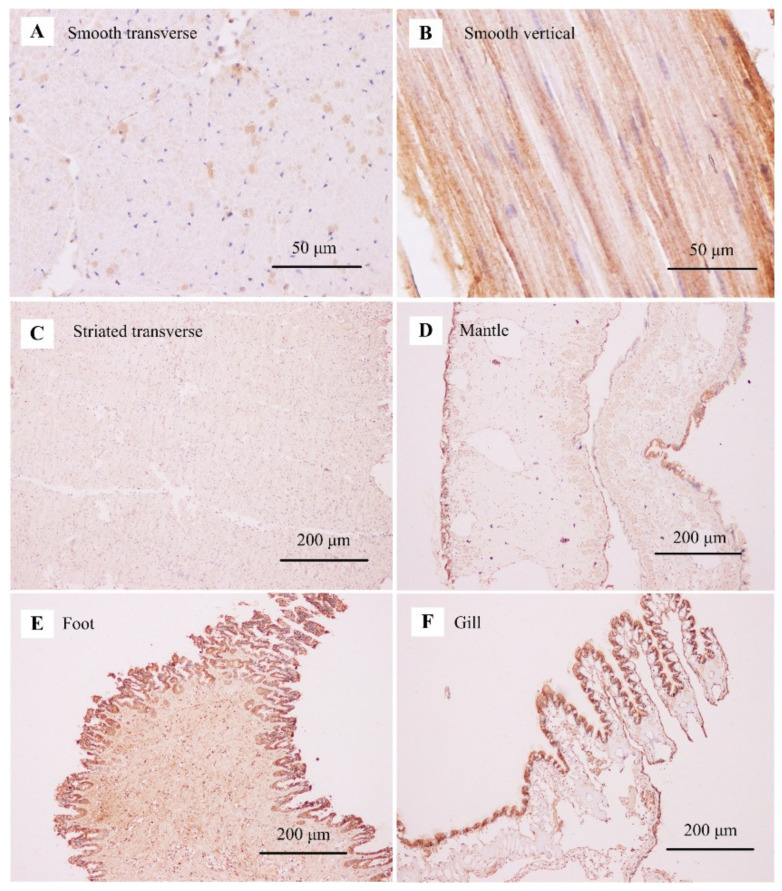
The immunohistochemistry staining for tissue-specific protein expression pattern of paramyosin. (**A**) Smooth adductor muscle in a transverse section; (**B**) smooth adductor muscle in a vertical section; (**C**) striated adductor muscle in a transverse section; (**D**) mantle tissue; (**E**) foot; and (**F**) gill.

**Figure 8 biology-11-00453-f008:**
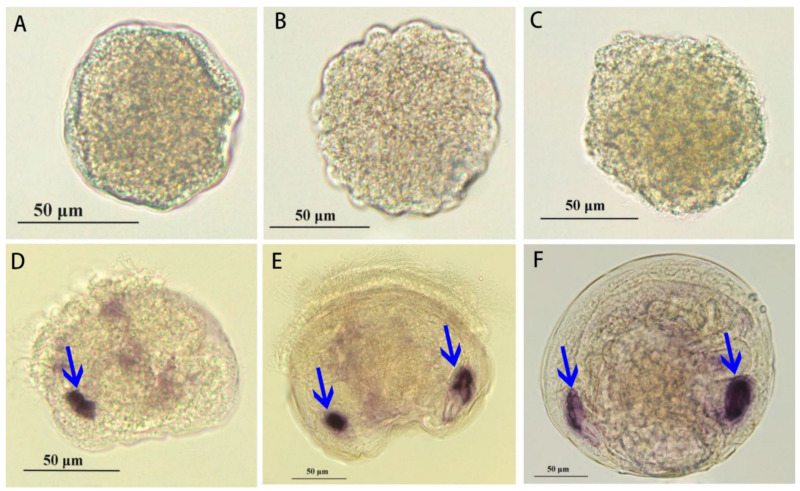
The spatial expression of paramyosin mRNA at different stages determined by whole- mount in situ hybridization. (**A**) multicellular stage; (**B**) blastula; (**C**) trochophore; (**D**) D-veliger stage; (**E**) umbo; and (**F**) spat. Scale bar = 50 μm.

**Table 1 biology-11-00453-t001:** The basic information on primers used in the experiment.

Primer Name	Sequences	Product Size(bp)	Information
paramyosin-full_F1	AGCTCCATTTCGTGATTTCCT	3763	PCR
paramyosin-full_R1	CACGTGTCCTCCATACAAACA		PCR
Paramyosin-F	ACATTACAAGGGCTAGTATTTAAAGCATTCGT	1520	PCR
Paramyosin-R	CTGTCTGTTCCTCTTGGTGAGATCC		PCR
Paramyosin_qPCR_F	GAGTCTGTCAGGAGGAAGTTT	158	qPCR
Paramyosin_qPCR_R	ACGATGATGGTGAGGTTTT		qPCR
Paramyosin_F	TGGATGACGAGTCTAGGCAA	611	in situ hybridization
Paramyosin_R	GGCTTTGTTGGCTTCCTTGA		in situ hybridization
wish_Paramyosin_F	GATCACTAATACGACTCACTATAGGGTGGATGACGAGTCTAGGCAA	611	in situ hybridization
wish_Paramyosin_R	GATCACTAATACGACTCACTATAGGGGGCTTTGTTGGCTTCCTTGA		in situ hybridization
UBQ_F	TCGCTGTAGTCTCCAGGATTGC	184	internal control
UBQ_R	TCGCCACATACCCTCCCAC		internal control

**Table 2 biology-11-00453-t002:** The phosphorylation sites identified in paramyosin of *Patinopecten yessoensis*.

No.	Position	Residue	Sequence Context	Score	Kinase	Positive Prediction
1	17	T	HVKRTYQVY	0.65	PKC	YES
2	28	T	TSPTTQNRF	0.53	DNAPK	YES
3	39	S	TGTISRRAS	0.80	PKC	YES
4	49	S	SQIQSGAIA	0.59	PKC	YES
5	56	S	IASSSSQLR	0.60	PKC	YES
6	61	T	SQLRTSSSQ	0.54	PKC	YES
7	64	S	RTSSSQVKR	0.75	PKC	YES
8	69	T	QVKRTIISS	0.66	PKC	YES
9	74	S	IISSSSSAA	0.74	PKC	YES
10	80	S	SAAQSSSNV	0.52	cdc2	YES
11	82	S	AQSSSNVNV	0.60	cdc2	YES
12	155	T	GLQSTQTDI	0.59	DNAPK	YES
13	157	T	QSTQTDIAR	0.88	PKC	YES
14	181	Y	ANAQYEAQE	0.50	EGFR	YES
15	187	S	AQESSQRKR	0.69	PKC	YES
16	233	S	MANDSLAKA	0.56	PKC	YES
17	240	S	KAKSSLDMR	0.71	PKA	YES
18	251	S	GLESSNIRL	0.52	cdc2	YES
19	362	S	SKANSDLVA	0.58	cdc2	YES
20	418	T	KAKLTNEVR	0.53	PKC	YES
21	439	T	VQDLTKRNR	0.87	PKC	YES
22	468	S	AALRSEKAA	0.71	PKC	YES
23	535	S	AERDSLAAA	0.63	PKA	YES
24	609	S	KSEISRIKK	0.71	PKC	YES
25	617	T	KKYETDIRE	0.57	CKII	YES
26	642	S	KQIRSLHIR	0.53	PKA	YES
27	701	T	KNAETELNE	0.56	CKII	YES
28	736	T	SAMQTDLDE	0.66	CKII	YES
29	893	Y	TMSKYRKAQ	0.52	EGFR	YES
30	915	T	EKNLTHTRR	0.71	PKC	YES
31	917	T	NLTHTRRTR	0.76	PKC	YES
32	920	T	HTRRTRSMS	0.65	PKC	YES

## Data Availability

The sequences obtained and analyzed during this study were deposited in the GenBank database under the accession numbers OL331027.

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
