# Peer review of "Molecular Characterization and Expression Pattern of Paramyosin in Larvae and Adults of Yesso Scallop"

_biology, 2022, doi:10.3390/biology11030453_

Round 1
Reviewer 1 Report
Xiujun Sun and his colleagues analyzed the characterization of paramyosin in Yesso Scallop Patinopecten yessoensis, which will help uncover the potential function of myofibrillar proteins in Molluscs. The work represents a big volume of work, the analysis is well performed and the manuscript is easy to read.
Here is a minor tip:
- Figure 4 shows that in the N-terminal region, more than 50 amino acids appeared to be the unique sequences of the two scallops. Can the author briefly discuss the influence of differences in amino acid sequences (or gene sequences) on protein structure and function?
Author Response
Response to Reviewer 1 Comments
Point 1: “Figure 4 shows that in the N-terminal region, more than 50 amino acids appeared to be the unique sequences of the two scallops. Can the author briefly discuss the influence of differences in amino acid sequences (or gene sequences) on protein structure and function?”.
Response 1: As suggested, we have added the related discussion on the influence of differences in amino acid sequences (or gene sequences) on protein structure and function in section 4.1. The detailed description has been provided in the main text.

Reviewer 2 Report
The authors report the results of Patinopecten yessoensis paramyosin analysis. There is original information concerning the expression of this protein in this particular mollusk.
The introduction does not include relevant information, furthermore it is repetitive on basical material.
The authors fail to expose the importance of the study of this protein in terms of the analysis that they report, considering the citation they made to the research in Patinopecten yessoensis that they have already published.
The coding sequence for paramyosin was submitted to GenBank registration, which is one of the original findings for publication. They included theoretical predictions for protein structure and phosphorylation sites; the discussion for the kinases that can be modulating the fuction of paramyosin in the smooth adductor muscle needs more information.
They conclude on the uniqueness for the phosphorylation sites of paramyosin, although, the figure 4 includes the 92.83% similar sequence of P. maximus, which has also some of these consesus sites; it is also needed a color code for this figure.
The discussion of the developmental expression of paramyosin can be improved in terms of the importance of the “well-organized” muscle system.
The manuscript needs english editing, although it might be some carelessness in writing it, considering some mistakes like “paramyosion”, “paraymosin”, “whole-amount”, etc.
Author Response
Response to Reviewer 2 Comments
Point 1: The authors report the results of Patinopecten yessoensis paramyosin analysis. There is original information concerning the expression of this protein in this particular mollusk. The introduction does not include relevant information, furthermore it is repetitive on basical material.
Response 1: As suggested, we have added the relevant information and revised the Introduction section. The detailed description is provided in the main text.
Point 2: “The authors fail to expose the importance of the study of this protein in terms of the analysis that they report, considering the citation they made to the research in Patinopecten yessoensis that they have already published.”
Response 2: We have revised the Introduction section according to the suggestion. We have provided the importance of this study for molecular characterization of paramyosin in scallops.
Point 3: The coding sequence for paramyosin was submitted to GenBank registration, which is one of the original findings for publication. They included theoretical predictions for protein structure and phosphorylation sites; the discussion for the kinases that can be modulating the fuction of paramyosin in the smooth adductor muscle needs more information.
Response 3: As suggested, we added some discussion of the kinases in section 4.4. The detailed description has been provided in the main text.
Point 4: They conclude on the uniqueness for the phosphorylation sites of paramyosin, although, the figure 4 includes the 92.83% similar sequence of P. maximus, which has also some of these consesus sites; it is also needed a color code for this figure.
Response 4: As suggested, we added the color boxes in Figure 4.
Point 5: The discussion of the developmental expression of paramyosin can be improved in terms of the importance of the “well-organized” muscle system.
Response 5: As suggested, we have improved our discussion on the importance of the “well-organized” muscle system in section 4.3,. The detailed description has been provided in the main text.
Point 6: The manuscript needs english editing, although it might be some carelessness in writing it, considering some mistakes like “paramyosion”, “paraymosin”, “whole-amount”, etc.
Response 6: As suggested, we have revised our MS by correcting the wrong spelling and grammar errors.
Reviewer 3 Report
The paper titled “Characterization of paramyosin in Yesso scallop Patinopecten yessonensis reveals its potential phosphorylation sites” reported the cloning and characterizing the full-length paramyosin from Patinopecten yessonensis using WISH, immunolocalization, as well as sequence analysis and prediction tools. The experiments are clearly described, and the conclusion is drawn carefully. However, I would suggest the authors consider changing the title since there are no functionality experiments besides computational prediction presented regarding phosphorylation. Plus, it seems to me that finding the phosphorylation sites in paramyosin of mollusks is not a rare event [references 9,16,20,21]. Also, only 10% of references are recent (5 out of 47 after yr 2019), among them 60% are by the same group, which made me suspect the importance and novelty of the subject. Other points that need clarification are as follows.
- Line19, “the strong mRNA” to “strong paramyosin mRNA”
- Line146-148, please confirm if the names of the primers are used correctly.
- Line169, what is “the developed primary antibody”? please provide a citation or describe the origin of the antibody, also, it is important to reveal if the paramyosin antibody cross-react with myosin.
- Line177, I didn’t find a record by searching all data at NCBI website (https://www.ncbi.nlm.nih.gov/search/all/?term=OL33102) with the accession# provided
- Line344, discussion point 4.4, would the myosin and paramyosin interaction play a role in the mollusk catch mechanism?
Author Response
Response to Reviewer 3 Comments
Point 1: The paper titled “Characterization of paramyosin in Yesso scallop Patinopecten yessonensis reveals its potential phosphorylation sites” reported the cloning and characterizing the full-length paramyosin from Patinopecten yessonensis using WISH, immunolocalization, as well as sequence analysis and prediction tools. The experiments are clearly described, and the conclusion is drawn carefully. However, I would suggest the authors consider changing the title since there are no functionality experiments besides computational prediction presented regarding phosphorylation. Plus, it seems to me that finding the phosphorylation sites in paramyosin of mollusks is not a rare event [references 9,16,20,21]. Also, only 10% of references are recent (5 out of 47 after yr 2019), among them 60% are by the same group, which made me suspect the importance and novelty of the subject. Other points that need clarification are as follows.
Response 1: As suggested, we have changed the title to “Molecular characterization and expression pattern of paramyosin in larvae and adults of Yesso scallop”. We have added several relevant new literatures on paramyosin studies in recent years. However, the recent research on molecular characterization of paramyosin is very limited.
Point 2:” Line19, “the strong mRNA” to “strong paramyosin mRNA”
Response 2:. As suggested, we have revised to “strong paramyosin mRNA”.
Point 3: “Line146-148, please confirm if the names of the primers are used correctly.”
Response 3: As suggested, we have modified the primer names.
Point 4:“Line169, what is “the developed primary antibody”? please provide a citation or describe the origin of the antibody, also, it is important to reveal if the paramyosin antibody cross-react with myosin.”
Response 4: The present study raised the polyclonal antibody in rabbits against a synthetic peptide corresponding to a unique region of P. yessoensis paramyosin, with no cross-reactive with myosin.
Point 5: “Line177, I didn’t find a record by searching all data at NCBI website (https://www.ncbi.nlm.nih.gov/search/all/?term=OL33102) with the accession# provided.”
Response 5: As suggested, the sequence has been submitted to Genbank (Accession Number. OL331027) and will be published on Nov 1, 2022.
Point 6: “Line344, discussion point 4.4, would the myosin and paramyosin interaction play a role in the mollusk catch mechanism?”
Response 6: As suggested, myosin and paramyosin interaction play a role in the mollusk catch mechanism. Paramyosin and myosin rod shared similar structural feature, which could be critical for these two proteins to assemble together in the thick filaments. The possible interaction of paramyosin with myosin may impair the association of the myosin heads with F-actin, suggesting that paramyosin may serve as part of a relaxing mechanism within invertebrate muscles. More and more evidences indicate that paramyosin phosphorylation is essential for muscle force generation in invertebrate muscles, probably involved in catch contractions. The interaction of the entire rod region of myosin with paramyosin core might be modulated by phosphorylation of paramyosin during the catch state. We have provided the detailed description in the main text.
Round 2
Reviewer 2 Report
The authors included information in introduction and discussion of the results. They have considered the parity in P. yessoensis and P. maximus paramyosin sequences, and pointed out the phosphorylation sites marked with blue boxes, but they still concluded on the uniqueness for P. yessoensis. The have revised the inaccuracies in description of methods, although the english revision is still needed, in particular for the use of some verbs.
Author Response
Point 1: The authors included information in introduction and discussion of the results. They have considered the parity in P. yessoensis and P. maximus paramyosin sequences, and pointed out the phosphorylation sites marked with blue boxes, but they still concluded on the uniqueness for P. yessoensis. The have revised the inaccuracies in description of methods, although the english revision is still needed, in particular for the use of some verbs.
Response 1: As suggested, we have revised our conclusion on the unique phosphorylation sites in scallops. According to the suggestion, we have our manuscript edited by a native English-speaking colleague, in particular for the use of some verbs.
Reviewer 3 Report
I appreciate the authors’ efforts to improve the manuscript. Here is the only issue needs clarification: Line 178-181: about the rabbit anti-paramyosin antibody. Although the authors had added some details regarding the lab-generated antibody, it is not clear if this antibody was from others or has been described in a published paper. If it is the first time the antibody was produced in the lab, a session in Method should be added including IRB approval of animals use. Please either provide the citation or revise the Methods.
Author Response
Point 1: I appreciate the authors’ efforts to improve the manuscript. Here is the only issue needs clarification: Line 178-181: about the rabbit anti-paramyosin antibody. Although the authors had added some details regarding the lab-generated antibody, it is not clear if this antibody was from others or has been described in a published paper. If it is the first time the antibody was produced in the lab, a session in Method should be added including IRB approval of animals use. Please either provide the citation or revise the Methods.
Response 1: In the present study, the polyclonal antibody of paramyosin was developed in the laboratory of Antibody Service Department of Shanghai Sangon (China), which were carried out under the Regulations of Hubei Province on the Administration of Experimental Animals (License key: SCXK2019-0023). As suggested, we have revised the description in the Methods.